# Modeling Paratuberculosis in Laboratory Animals, Cells, or Tissues: A Focus on Their Applications for Pathogenesis, Diagnosis, Vaccines, and Therapy Studies

**DOI:** 10.3390/ani13223553

**Published:** 2023-11-17

**Authors:** Ana Jolly, Bárbara Fernández, Silvia Leonor Mundo, Natalia Elguezabal

**Affiliations:** 1Cátedra de Inmunología, Facultad de Ciencias Veterinarias, Universidad de Buenos Aires, Av. Chorroarín 280, Buenos Aires C1427CWO, Argentina; bfernandez@fvet.uba.ar (B.F.); smundo@fvet.uba.ar (S.L.M.); 2Instituto de Investigaciones en Producción Animal (INPA), CONICET-Universidad de Buenos Aires, Av. Chorroarín 280, Buenos Aires C1427CWO, Argentina; 3Instituto de Investigación y Tecnología en Reproducción Animal (INITRA), Facultad de Ciencias Veterinarias, Universidad de Buenos Aires, Av. Chorroarín 280, Buenos Aires C1427CWO, Argentina; 4Departamento de Sanidad Animal, NEIKER-Instituto Vasco de Investigación y Desarrollo Agrario-Basque Research and Technology Alliance (BRTA), 48160 Derio, Spain

**Keywords:** paratuberculosis, in vivo, in vitro, ex vivo, infection model, macrophage, neutrophil, epithelial cell, rabbit, experimental infection

## Abstract

**Simple Summary:**

Paratuberculosis is a chronic infectious disease that affects a wide variety of domestic and wild animals. It is considered one of the diseases with the highest economic impact on the ruminant industry. Despite many efforts and intensive research, the existing measures for paratuberculosis control are still insufficient. Experimental models are a fundamental tool for making progress in the knowledge and control of this disease. Here, we review the potential and limitations of the different experimental approaches currently used in paratuberculosis research, focusing on laboratory animals and cell-based models.

**Abstract:**

Paratuberculosis is a chronic granulomatous enteritis caused by *Mycobacterium avium* subsp. *Paratuberculosis* that affects a wide variety of domestic and wild animals. It is considered one of the diseases with the highest economic impact on the ruminant industry. Despite many efforts and intensive research, paratuberculosis control still remains controversial, and the existing diagnostic and immunoprophylactic tools have great limitations. Thus, models play a crucial role in understanding the pathogenesis of infection and disease, and in testing novel vaccine candidates. Ruminant animal models can be restricted by several reasons, related to space requirements, the cost of the animals, and the maintenance of the facilities. Therefore, we review the potential and limitations of the different experimental approaches currently used in paratuberculosis research, focusing on laboratory animals and cell-based models. The aim of this review is to offer a vision of the models that have been used, and what has been achieved or discovered with each one, so that the reader can choose the best model to answer their scientific questions and prove their hypotheses. Also, we bring forward new approaches that we consider worth exploring in the near future.

## 1. Introduction

Johne’s disease or paratuberculosis (PTB), caused by the *Mycobacterium avium* subspecies *paratuberculosis* (MAP), is a chronic granulomatous enteritis which affects large populations of ruminants globally. The primary route of MAP transmission is the ingestion of colostrum, milk, food, or water contaminated with MAP. Following transcytosis through the gastrointestinal barrier, MAP is taken up by the resident macrophages and dendritic cells present in the subepithelial lamina propria. In this cellular niche, MAP survives and replicates, triggering granulomatous inflammation. Only a small proportion of chronically infected animals will develop a fatal, progressive form of the disease. However, subclinical PTB can negatively affect weight gain, milk production, and fertility, and can also lead to increased somatic cell counts, incidences of clinical mastitis, and susceptibility to other diseases. Prevention strategies to combat the spread of PTB among herds involve adhering to strict calving practices to ensure that young, susceptible animals do not come into contact with MAP-contaminated colostrum, milk, or feces. Achieving PTB control requires critical advances in both diagnosis and vaccination areas. It is also well accepted that progress in these areas requires a deeper knowledge of host–pathogen interactions during infection. Although PTB was first described in the late 1800s, its pathogenic mechanisms have not been clearly defined. This is probably due to the lack of economic and adequate laboratory animals that faithfully resemble ruminant PTB, and also to the absence of harmonization in the use of models among PTB research groups. It has been said that the optimization of an animal model with which to study MAP infection has been stymied by the complexity of this disease [1]. For this reason, some groups have focused on large animal models, as these are the target species for which control is desired. However, these animal models are, many times, overlooked by other research groups, in part because of the prohibitive costs of maintaining these animals in a research environment.

Extensive reviews have been previously published elsewhere describing both ruminant and laboratory animal models [2,3]. Here, we review the potential and limitations of the different experimental approaches used in PTB research, focusing on non-ruminant animals and cell-based models. Figure 1 summarizes this information and illustrates the relative contribution of each model to this field. In addition, we complete the review by bringing forward new approaches that we consider worth exploring in the near future.

## 2. In Vivo Animal Models in PTB Research

As previously stated, ruminant species are the first choice for study, considering their role as natural hosts of this bacterium and because the disease causes economic losses mostly in ruminant livestock, meaning that a vaccine for these species would be highly valued. However, the use of ruminant species as models, even small ones, may be restricted by several reasons related to space requirements, the cost of the animals and the maintenance of the facilities, the volume of daily excreta produced and, therefore, the need for permanent cleaning, the complexity of restraint maneuvers, among others. In addition, when the observation of clinical features is pursued, ruminant MAP infection becomes lengthy, adding to the cost and slowing down scientific progress. Laboratory animal models are normally the more realistic choice when the evaluation of new therapies, vaccines, or pathogenesis studies is the goal. Although it is impossible to reflect the complete image of PTB in laboratory animals, these non-ruminant models are advantageous because they are less expensive and, in some cases, PTB-compatible signs can be achieved in a shorter period of time. They can be used in the first steps of vaccine screening or when large animal modeling is not possible. In this review, we will focus on other alternative in vivo models, based on laboratory species. Also, we will refer to the amoebae as the simplest eukaryotic model of infection.

### 2.1. Mouse Model

Mice have been extensively used in PTB research for pathogenesis and vaccine prototype testing. Actually, the use of this model has been recommended for the initial steps of vaccine screening due to the similarities of granulomatous inflammation and immunological responses between mice and ruminants [2]. An in-depth review of the studies that included this animal model was performed recently [3]. Therefore, this section will only highlight the main features of this model.

Mice strains with different genetic backgrounds have been assayed, although BALB/c and C57BL/6 have been the most widely used strains [3]. MAP was administered through the intraperitoneal route (IP) in most cases (67% of the studies reviewed in [3]), followed by the oral route (20% of the studies reviewed in [3]). The oral route has been shown to be less reproducible, and it does not recreate all the classical features of intestinal PTB [3,4], only producing granulomatous lesions in the mesenteric lymph nodes [5] and disseminated infection in high-dose challenges [4]. However, the immune profile achieved with this model is similar to that observed in cattle [4]. The intraperitoneal challenge produces changes by 6–12-weeks postinfection (w.p.i.), mainly a lymphocytic inflammatory response, whereas a granulomatous inflammation is detected by 12 w.p.i. [6]. Shao et al. [7] reported differences in the CFU load in the liver from 2 to 8 w.p.i with the intraperitoneal route, with no bacterial recovery after that. They also detected differences in weight gain among the groups from 2 to 12 w.p.i. Also, a recent study compared the early stage of MAP infection among the different administration routes in the murine model, and detected increases in the weight of the spleen and the liver, and more severe histopathological changes through the IP route compared to the oral route [8].

The infection outcome in the murine model is normally evaluated by assessing the MAP burden in the tissue and histopathology of the liver, spleen, and intestines, since gross pathological lesions are normally not achieved. The drawback of this model regarding PTB is that it fails to reproduce diarrhea or severe intestinal lesions. Blood extraction in mice can be difficult, impeding the in vivo follow-up at the same time of many of the parameters that are quantifiable in blood or its components (humoral or cellular). Performing ex vivo functional cell assays can be accomplished by increasing the number of animals, dedicating each individual to a specific time-point, and by isolating bone marrow at the experiment endpoints when an immune cell analysis is required [9,10,11].

### 2.2. Rabbit Model

Rabbits are susceptible to the development of MAP infection under both experimental [12,13,14,15,16] and natural [17,18] conditions. Diarrhea, fecal shedding, weight loss, and intestinal colonization signs can be studied with the rabbit model.

The first studies on the experimental MAP challenge in rabbits were mostly carried out to prove infection, and to attempt to develop an appropriate model with which to study its pathogenesis and screen vaccine candidates. Therefore, different animal ages; challenge doses; and routes, including intraperitoneal, intravenous, and oral, were tested [13,15,19,20,21]. The oral route, mimicking the natural route of MAP infection, was the most pursued. Further studies have used this challenge route to evaluate the impact of diet [16] or a probiotic [22] on MAP infection, MAP infection on the microbiome [23], the vaccination sequence on the progression of MAP infection and the immunological response [24], and the passively transferred antibodies on MAP colonization [25]. This oral challenge model has also been complemented with ex vivo assays that have permitted the functional analysis of immune cells and the study of trained immunity exerted by vaccination [26]. When using this model, it should be considered that the reliable reproduction of the typical clinical, pathological, and microbiological signs seems to depend greatly on the strain’s origin and passage, obtaining higher degrees of infection when low-passage cattle field strains are used. The key aspects of the most relevant studies involving the experimental, oral MAP infection in rabbits are summarized in Table 1.

The rabbit digestive tract can closely resemble that of ruminants because of the similarities between the *sacculus rotundus* in rabbits and the ileocecal valve in ruminants. Also relevant, to some extent, is the fact that the physiological body temperature of the rabbit (38.3–39.4 °C) is closer to that of the cow (37.8–39.2 °C) compared to that of the mouse (36.5–38 °C). This could somehow impact the results, since it seems that MAP infection is enhanced at 39 °C compared to 37 °C [28]. Finally, the rabbit’s biological cycle allows for short-term challenge experiments permitting the development of mild PTB signs in less than 5 months. These characteristics, together with their size, handling, and ease of blood sampling, as well as the volume of blood it is possible to collect in the periodical sampling, make rabbits a convenient experimental species with which to model PTB, offering important advantages for studying key aspects of paratuberculosis that cannot be investigated with other laboratory animals because many symptoms, such as diarrhea or fecal shedding, cannot be reproduced in those models. Regarding their immunological system, two particular features place the rabbit as a model closer to ruminants than the mouse: on one hand, the presence of a primary intestine-associated lymphoid structure, which is a determinant for the generation of diversity in the B-cell repertoire (ileal Peyer’s patch in ruminants, and cecum in rabbits); and, on the other hand, the fact that both rabbits and ruminants have been classified as a γδ high species according to the much higher proportion of γδ T lymphocytes (up to 60%) in their T-cell circulating pool than that of the γδ low species, such as the mouse, in which the proportion of γδ T cells is around 5% [29]. Although the role of γδ T lymphocytes in PTB is not fully understood, bovine γδ T lymphocytes are known to be early responders to MAP infection [30,31], and evidence for the key role of this subpopulation in the clinical phase of the disease has also been found from observations of naturally infected animals [32,33,34].

### 2.3. Ferret Model

The ferret has been used as a model for infectious diseases [35], mainly for tuberculosis (TB), for which this species has been postulated as an environmental reservoir [36,37]. In 1997, Lugton and coworkers [38] reported on the chronic, low-level natural infection of ferrets with *M. avium* subsp. *avium*. More recently, Bannantine et al. [39] published the first report using ferrets as an experimental model for MAP infection. They stated that 3–4-month-old female ferrets challenged orally with 10^8^ CFU MAP lost body weight after 16 w.p.i., and developed a specific antibody response by week 13, while the IFN-γ response remained undetectable until 20 w.p.i., the final endpoint of their experiment. Taking into account these results, they proposed ferrets as a valid small animal model for testing the virulence of MAP strains. As ferrets have a very short intestinal tract, an adaptation due to their strict carnivore diet [40], Bannantine et al. [39] suggested that this anatomical feature could be an advantage for using this species as an experimental model for PTB, as the site of infection after oral inoculation would be well circumscribed. However, the usefulness of this carnivorous animal as a model for the study of PTB should be further confirmed, since it is a monogastric species that has not been shown to be susceptible in nature, there is only the one study, and the lack of species-specific immunological reagents, along with the outbred nature of the model, are major drawbacks.

### 2.4. Protozoan Model

Eukaryotic organisms like amoeba have been noted to interact with mycobacterial species in the environment. Based on the ability of some mycobacteria to grow within environmental amoebae [41,42,43], and the data suggesting the increased virulence of mycobacteria upon replication within amoebae [44,45], Phillips et al. [46] evaluated the influence of MAP infection on amoeba metabolic activity in the *Acanthamoeba castellanii* infection model, and found that this amoeba could be used as a quick initial screening tool for the selection of virulence factors with relevance to macrophage infection. The main limitation of this model is that it is a unicellular organism, and the utility of this model would seem to be limited to virulence factors research studies.

## 3. Ex Vivo and In Vitro Systems 

Apart from in vivo models, most of the contributions to the knowledge of MAP–host interactions have been made from work on ex vivo or in vitro models, focused on the use of cells of epithelial and/or of phagocytic origin, since the target organ of this infection in ruminants is the intestine and the associated lymphoid tissue, and macrophages are the main cellular niche in which MAP can survive and replicate within its host. 

Ex vivo and in vitro cellular infection assays can be fast and cost-effective alternatives to in vivo experimentation for the evaluation of MAP–host cell interactions. These assays can contribute information regarding pathogenesis and an initial notion of vaccine effects. In any case, the most relevant models found in the literature are further described in the following sections.

### 3.1. Ex Vivo Models

Ex vivo means “outside of a living body”, implying that cells or tissue have been previously isolated from an animal and maintained in culture conditions for a study. Therefore, an in vitro assay involves an established cell system, such as a cell line, whereas an ex vivo assay is based on cells or tissue from a living organism, that is, primary cells. In the same way, the level of cellular complexity is superior in an ex vivo assay, being closer to the organism’s natural conditions and, therefore, offering advantages. Although higher variability is observed due to the use of more than one individual, it is closer to reality, especially regarding PTB, a disease in which some animals are susceptible to the infection and develop clinical signs, while others are resilient and remain asymptomatic long term.

These models can be based on cells isolated from tissues and/or whole blood, or on tissue slices from different organs, conserving the original cell structures and intercellular connections. In all cases, treated or untreated, healthy or diseased animals can be used as donors, increasing the complexity of the resulting information from in vivo experiments. In PTB research, different ex vivo models have been used.

#### 3.1.1. Intestinal Models

Ex vivo models using intact intestinal tissue have been developed, such as the culture of intestinal explants obtained with a biopsy, first described in 1959 [47], and the everted sleeve method [48]. These models were first developed from mouse samples, and then from rat, pig, and human explants as well. The use of both models has been explored in the context of PTB research: Schleig and colleagues [49] showed that explant cultures of different intestinal sections are suitable models with which to study the early events of MAP invasion, such as the cellular attachment process. In the same year, Sigurdardottir and colleagues demonstrated that MAP invasion occurs in small intestinal areas, with and without Peyer’s patches, using the everted sleeve method [50]. More recently, it was described that intestinal explants can be stimulated ex vivo, inducing immune responses to a wide range of stimuli and detecting changes in expressed cellular phenotypes, as well as in the levels of biomarkers of inflammation [51]. We believe that primary cultures obtained from animals with PTB and different degrees/types of intestinal pathology versus healthy control animals could be useful for studying the differences in MAP pathogenesis.

The simplest intestinal model could be considered to be the primary culture of intestinal epithelial cells. Primary cultures of bovine epithelial cells from the small or large intestines (BIEC) have been used to evaluate the effects of the virulence factors of other microbes, toxic compounds, and the innate immune responses through Pattern Recognition Receptor (PRR) signaling. Li et al. [52] used BIEC obtained from neonatal bovine ileum to evaluate the effect of *Saccharomyces cerevisiae* components on MAP adherence. However, these cell suspensions might be contaminated with non-epithelial cells (mostly fibroblasts) and, therefore, a series of purification steps would be required to obtain relatively pure BIEC [53]. The main constraints in culturing primary cells are that they normally enter into a non-dividing state after a few passages and, also, the possible variability between donors. However, this system would be valuable for evaluating MAP intestinal invasion and early pathogenesis. 

#### 3.1.2. Immune Cell Models

Macrophages are considered key players in PTB since these immune cells act as reservoirs providing a favorable environment for MAP replication. Once phagocytosed by macrophages, MAP is able to arrest phagosome–lysosome fusion and, therefore, inhibiting the acidification of the cellular compartment and hampering pathogen destruction [54]. MAP can control monocyte and macrophage apoptosis, thereby feeding the infection cycle [55]. The other phagocytes that have been less studied, probably because they have not been considered as important cell types in MAP pathogenesis, are neutrophils. 

##### Macrophages and Monocytes

Macrophages play a critical role in the host–pathogen interaction of PTB. Actually, they have a dual role, being effector cells that can mediate both the destruction of MAP and its survival, proliferation, and dissemination after the pathogen inhibits phagolysosome maturation. Monocyte-derived macrophages (MDMs), generated from the peripheral blood mononuclear cells (PBMCs), have been widely used to model the macrophages for ex vivo studies. As for PTB, MDMs have been extensively used to study MAP pathogenesis, screen for MAP mutants, study the immune response (phagocytosis, growth, and cytokine production) against MAP infection and, in functional assays, to aid in vaccine prototype evaluation or to complement specific resistance/susceptibility-related host trait findings.

MDMs have been infected with MAP alone, or in a co-culture with other cell types. In this section, we will discuss the studies involving MDMs cultured alone, whereas MDMs co-cultured with other primary cells or cell lines will be discussed in the co-culture section.

Regarding the methodology for monocyte isolation and enrichment, the majority of studies included protocols that isolated the PBMCs first, and selected for adherent cells on plastic culture plates for their subsequent differentiation into macrophages. A few works included protocols where MDMs were obtained after monocyte isolation from the PBMCs using an anti-CD14 antibody coupled to magnetic microbeads [56,57,58]. Arteche-Villasol and collaborators [58] found that this magnetic separation yielded a higher proportion of monocytes from the PBMC fraction than the plate method, although the latter achieved higher yields of MDMs than those isolated through the former in sheep and goats. It is worth mentioning that the monocyte isolation technique not only affects the yield and purity, but also impacts the resulting phenotype of the cultured cells [59]. Temperature is another important factor to consider, as it has been demonstrated that physiological temperature greatly influences MAP gene-expression profiles and the speed of macrophage invasion [28]. Many groups have further contributed methodologies that could aid in the evaluation and quantitation of bacterial intracellular growth, and that could be complementary to MAP infection assays in MDMs. Mitchell et al. [60] showed that real-time qPCR assays provide a more accurate and precise method for evaluating MAP intracellular growth dynamics in MDMs to study strain differences compared to fluorescent quantitation. Also, confocal microscopy has been suggested as a valuable tool with which to quantify MAP in MDM infection assays [61].

The first studies performed with electron microscopy showed that MAP had not degraded four weeks after infecting MDMs [62], indicating that MAP could actually survive inside macrophages. Also demonstrated was the fact that both monocytes and MDMs required the presence of serum in order to efficiently phagocytose MAP, and that MDMs showed higher phagocytosis levels than freshly adherent monocytes [63]. Later on, Khalifeh and collaborators found that IL-10 and TGF-β inhibited the destruction of intracellular MAP in the presence of IFN-γ in the MDMs from naturally infected cows, compared to the MDMs from healthy animals, suggesting important immune-regulatory roles for these cytokines during an infection with MAP [33]. Afterwards, Weiss and collaborators, using a combination of gene expression studies and functional assays, suggested that the inhibition of phagolysosomal function and apoptosis, and the expression of the inhibitors of macrophage activation, were probably important factors behind the survival of MAP in bovine macrophages [64]. The same group later studied the role of IL-10, showing that the neutralization of this cytokine enabled the macrophages to kill 57% of the MAP organisms within 96 h, along with an increase in the acidification of the phagosomes, the apoptosis of the macrophages, and the production of nitric oxide (NO), suggesting that the induction of IL-10 expression by MAP could be considered a virulence factor [65]. All these studies were important for gaining knowledge of the MAP–macrophage interaction and the role of IL-10 in PTB and, ultimately, set the conditions for future assays.

Also regarding pathogenesis elucidation, Khalifeh and coworkers evaluated the role of NO production in MAP viability within macrophages, by culturing MDMs from subclinically or clinically infected cows and healthy controls, and observed that NO production was highest in the subclinically infected cows, and that it peaked when the cells were co-stimulated with MAP and IFN-γ [66]. The authors hypothesized that in the transition from the subclinical to clinical stage, an increase in the bacterial load of the cells ceased IFN-γ upregulation, and led to an impairment of NO generation.

MDM ex vivo assays have also been used to generate knowledge for novel vaccine and therapy design, either by gaining knowledge of cell–MAP interactions at the molecular level, or by treating animals and performing ex vivo functional assays for evaluation afterwards. In many studies, it has been shown that the inhibitors of mitogen-activated protein kinases (MAPK), such as MAPKp38, increase the production of nitric oxide after a MAP infection [67], that vitamin D3 analogs promote the phagocytosis of MAP by MDMs [68], and that the antibodies directed against lipoarabinomannan (LAM) used to opsonize MAP prior to MDM contact increase macrophage apoptosis and TNF-α secretion [69]. Vaccination has been shown to enhance the microbicidal activity of macrophages against MAP ex vivo, as seen by a significant reduction in MAP viability (CFU), as well as the upregulation of iNOS and IL-10 by RTqPCR in infected MDMs from vaccinated animals, compared to non-vaccinated animals [70]. All these studies have shown that different treatments can enhance macrophage functions, at least in ex vivo conditions.

Studies focused on host genetics and MAP strain differences have also been performed using MDMs aimed at identifying markers that can be exploited for therapy or diagnosis. Actually, the possibility of finding strain differences among MDMs has been exploited to screen attenuated [57,71] and transposon mutants [72] in bovine MDMs in studies looking for novel vaccine candidates. Some studies have failed to find differences in functionality between the MDMs of the PTB-positive and control animals, but have found important strain differences [73], whereas other studies have shown that MDMs with certain SNPs (TLR2-1903 TT genotype) produce higher levels of IL-12p40 and IL-1β when stimulated with MAP, compared to cells derived from other genotypes (TLR2-1903 CT and CC), suggesting the use of these SNPs in marker-assisted breeding strategies as an additional tool in PTB control strategies [74]. The importance of host genetics has been further demonstrated in studies that have showed that the ability of MDMs to limit MAP viability is associated with specific genetic profiles [75]. Many studies have identified genetic variants in the regulatory pathways of the macrophages that may affect the susceptibility of cows that are healthy/resistant to MAP infection [76,77]. All these studies support the use of genomic and transcriptomic approaches to enable the identification of the markers associated particularly with susceptibility to MAP infection.

In summary, MDM models have been and are very useful for studying pathogenesis focused on the MAP–macrophage interplay. However, we must bear in mind that they are insufficient to provide accurate information on the events that take place during natural infection, as shown by the analysis of the MAP transcriptome from the tissue of naturally infected cows, or from in vitro-infected macrophages [78].

##### Granulomas

The granuloma has been considered a hallmark of mycobacterial infection, and the evaluation of the factors that affect its development and evolution is complex. A model for the in vivo dynamics of bovine paratuberculosis granulomas has been proposed by Koets et al. [79]. In an effort to simulate the earliest events in the immune response leading to tuberculous granuloma formation, a number of in vitro models have been developed (reviewed in [80]). In the field of PTB, the early interaction and coordination of macrophages upon MAP infection has been studied, starting from MDM in vitro cultures that were kept for 10 days in order to form granulomas [81,82]. In the first study, Multiplicity of Infections (MOIs) of 1:8, 1:16, and 1:33 were assayed, and found that the highest number of microgranulomas was achieved with the lowest MOI of 1:8 [81]. In the second study, the authors found that at MOIs of 1:2 and below, the macrophages displayed increased longevity compared to the uninfected cells, and formed clusters that secreted the pro-inflammatory cytokines necessary for a cell-mediated immune response [82]. While at higher MOIs, the viability of the host MDMs was negatively impacted and the intracellular MAP reproduced over the first five days of the infection.

Since maintaining granuloma structures in vitro over an extended period of time is still a limiting factor for this model [83], only the features of the early stage of granuloma formation can be studied. Although it has been proposed that granulomas are marked by the interaction of various immune cell types, which acquire different phenotypes through it, as is the case in the M1/M2 polarization of macrophages, to date, only small and simple granuloma-like structures based on MDMs have been evaluated as a model for PTB. Such a model, although still lacking the degree of realism and cellular complexity achieved in vivo, could be refined and provide, in the near future, information related to relevant aspects of PTB pathogenesis: the mechanisms involved in granuloma induction and maintenance, the activation of MAP latency genes [84], the role of different leukocyte subpopulations, and the characteristics of the granulomas that favor bactericidal functions versus bacterial persistence [85].

##### Neutrophils

Several studies have shown that neutrophils can play an important role in innate immune protection against mycobacterial infections [86,87]. Although neutrophils are absent in the lesions of the advanced stages of MAP infection, their presence at infection sites in the early phases has been reported [88]. Furthermore, neutrophils seem to assist in the induction of specific Th1 and Th17 cells in response to a tuberculosis vaccine [89]. In any case, their possible role in defending against MAP has been evidenced in transcriptomic studies that have described the impairment of neutrophil recruitment and activation during PTB [90,91,92]. These findings have recently directed the focus onto neutrophils, leading to studies that have looked into the behavior of these immune cells against MAP in ex vivo assays.

Neutrophils kill microbes through a set of mechanisms that include phagocytosis, reactive oxygen species (ROS) production, degranulation, and extracellular trap (ET) release [93]. Neutrophils isolated from healthy cattle have shown ET liberation against MAP and the effective killing of these bacteria in vitro [94]. Caprine neutrophils have also demonstrated a strong innate response against MAP, using their entire repertoire of effector functions, ET release, degranulation, chemotaxis, and phagocytosis [95]. Vaccination against PTB in rabbits increased the phagocytic activity of these immune cells, along with their ability to release ET against mycobacterial and non-mycobacterial agents [26]. In this study, the degree of protection of the different vaccine prototypes was correlated to the ex vivo antimicrobial activity of the neutrophils, indicating that this cell type may play a role in protection.

Neutrophils are tricky cell types to work with, and functional assays must be performed with freshly isolated cells, making this model less attractive than monocytes or macrophages. The number of protocols for the isolation, culture, and evaluation of functionality is continuously growing, and are largely dependent on the host species [93]. An improvement of the methodology will probably nurture further research on this phagocytic cell and its role in PTB.

#### 3.1.3. Co-Cultures of Primary Cells Alone or Combined with Cell Lines

Co-cultures (a culture of more than one cell type) can be an option with which to study the interactions between different cell types. These kinds of models can give information about the crosstalk between the cell types evaluated and also about the host–pathogen interactions. Therefore, most ex vivo assays described in the literature have focused on macrophages, either alone or in co-culture with other cell types, including lymphocytes [96,97], PBMCs [98], neutrophils [94], or even epithelial cell lines [99,100]. The objective of each of these co-culture models varies depending on the cell type and on the origin of the cells (vaccinated, healthy, or MAP-infected animals).

Hostetter et al. [101] found that the mycobactericidal functions of MDMs were enhanced in MAP-infected cattle, and that the addition of autologous CD4+ T cells did not increase bacterial killing. However, CD4+ T cells from non-infected animals did increase bacterial killing in autologous macrophages. The same group tested the influence of γδ T cells on MDMs in vitro, and found that these cells from MAP-sensitized animals were not able to produce enough IFN-γ in order to enhance the mycobacterial killing or nitrite production by the infected macrophages [102].

Co-culturing MDMs with WC1+ and WC1− γδ T lymphocytes has shown that these lymphocyte subsets modulate the effector functions of MDMs, such as their MAP-killing ability [96]. Furthermore, Pooley and co-workers [97] developed a mycobacterial growth inhibition assay based on a monocyte–lymphocyte co-culture model, and used it to evaluate vaccination on sheep. They concluded that this system could quantify the ability of PBMCs to kill MAP, serving as a predictor of vaccine failure or non-response, and could be potentially useful for novel vaccine screening. Similarly, Davis and co-workers developed ex vivo platforms in cattle to study the functional activity of CD4+, CD8+, and γδ T and NK cells stimulated with MAP antigen-primed antigen-presenting cells (dendritic cells (DC) present in the blood, monocyte-derived DC, and monocyte-derived macrophages) [57,103,104,105]. The authors highlighted the usefulness of these platforms for assessing bacterial viability, the cells’ phenotype, cytotoxicity mechanisms and, consequently, for examining the potential role of the cell-mediated immune response in preventing the establishment of a persistent infection of MAP [106]. 

In the same way, another interesting ex vivo assay that complements an in vivo model is one based on the co-culture of MDMs with monocyte-deprived PBMCs that had been previously primed, or not, with MAP-stimulated DC. This assay showed that the macrophages from animals vaccinated with prototype vaccines, cultured alone or in a co-culture with unprimed PBMCs, showed an increased MAP-killing ability. This is in agreement with a more effective clearance of MAP infection from calf tissues in vivo [98].

The co-culture of MDMs with autologous neutrophils isolated from healthy cattle was performed to study the interaction of both cell types with MAP and *Mycobacterium bovis* alone, or in a co-culture, and showed their cooperation and synergistic effects against mycobacteria [94]. In this work, the co-cultures were performed by putting both cell types in contact in the same culture well or separately in a transwell system, sharing nutrients and metabolites but avoiding physical contact. The ET release and bacterial killing were evaluated along with cytokine secretion, offering information on the mechanisms that operate in the defense of these cell types against MAP.

Finally, using a permeable support system, Lamont and coworkers [99] were able to artificially construct an epithelial cell layer with bovine mammary epithelial cells (MAC-T cell line), along with the bovine MDMs immediately underneath them, to represent subepithelial macrophages. Thus, this system aimed at mimicking the MAP epithelial invasion a step prior to the uptake by the macrophages, and answered questions about the interaction between the bacteria and MAC-T cells, and the impact of this one on the macrophages [99,100]. This co-culture system showed that MAP leads to phagosome acidification in the epithelial cells and IL-β release that makes for an efficient epithelial transverse, ending in macrophage uptake [99]. Further studies using this model have permitted a deeper characterization of early MAP infection, identifying the metabolic, DNA repair, and virulence genes that could be considered as novel drug targets for pathogenesis studies [100].

#### 3.1.4. Organoids

Organoids are three-dimensional organ-like structures composed of functional, stem cell-derived multicellular aggregates that can be indefinitely propagated to resemble a specific organ when cultivated with essential niche factors. An intestinal organoid culture was first developed in mice and humans following the identification of the signaling pathways involved in the maintenance and proliferation of the fast-cycling intestinal stem cells that express high levels of leucine-rich repeat-containing G protein-coupled receptor 5 (Lgr5) [107]. In these species, organoids can be obtained from adult intestinal epithelial stem cells (Lgr5+ IESC), from adult reprogrammed stromal cells or induced pluripotent stem (iPSC) cells, or from embryonic stem cells (ESC). In the case of ruminants and other farm animals, although advances made recently in the field of the development of methodologies for obtaining ESC and iPSC [108,109], to the best of our knowledge, only methods for obtaining organoids from IESC have been described. Since the organoids are generated from stem cells, they have the capacity to generate crypt-like domains, with the potential to differentiate into many of the heterogeneous epithelial cell types, contrary to the transformed or immortalized cancer-derived cell lines, and spontaneously organize into tissue-like structures that reflect the characteristics of the digestive segment of origin [110]. The methodology for obtaining and working with enteroids is beyond the scope of this review, and can be consulted in the recent literature [111]. 

The protocols for culturing ruminant organoids from different digestive segments have been set out in [112,113,114,115,116,117,118,119,120]. Since the first report in 2009, intestinal organoids have evolved as a potential alternative to in vivo models for various experimental purposes, such as imaging, molecular analysis and gene editing, but also as reductionist approaches with which to study the interaction of epithelial cells with the other relevant actors in a physiological context, such as the immune cells [121,122] or microbiota [123,124], or even to evaluate the early host–pathogen interaction in the context of many infectious diseases (protozoa [125,126], virus [127], and bacteria [128,129,130,131,132,133]). 

Studies on animal organoids have gained more attention in various fields, including veterinary medicine, given recent advances in organoid technology (reviewed in [134]). In the field of PTB studies, the first steps in this direction are just beginning to be taken [119]. In that recently published work, a 3D bovine enteroid model was developed, with the apical side exposed on the exterior surface, enabling the infection of the organoids without the need for microinjection. It was concluded that this model was susceptible to infection with MAP and, therefore, useful for investigating the early stages of MAP pathogenesis. In the near future, this model could certainly be used to shed light on other aspects of pathogenesis, such as cell tropism, virulence factors, the biological differences among MAP genotypes, and exploring new signaling pathways targetable for disease prevention and, thus, facilitating the discovery of the interactions between the mycobacteria and host cells in a more physiological environment. Even so, these experimental systems still bear some limitations; namely, the lack of immune cells and other types of cells normally present in the architecture of tissue. So, the next generation of these organoids, includs co-cultures with the different cells of the immune system [122,135], could be very useful in the context of research into the immunopathogenesis of the infection and new vaccine candidate evaluation, by studying the fate of MAP infection in organoids co-cultured with immune cells, obtained from both vaccinated versus unvaccinated animals. Organoids co-cultured with macrophages could provide new evidence for the role of the M1/M2 dichotomy in the context of this infection. On the other hand, the genetic markers associated with the phenotypes of susceptibility or resistance to PTB have been described (reviewed in [136]), as well as the additional intermediate category of “tolerant hosts” [137]. Taking into account that organoids maintain an individual genetic fingerprint, the usefulness of this model for gaining knowledge of the internal differences between those phenotypes is questionable. 

Among the early events of MAP infection, the bacterial interaction of the MAP enterocytes, M cells [138,139,140], and goblet cells [88] on the lining of the epithelium has been identified. Similarly, organoids obtained from different intestinal segments have been used to study the regional differences detected in the invasion and response to MAP infection in ileum versus jejunum [141,142,143]. The use of intestinal organoids could be further used to deepen the knowledge of these MAP–cell interactions.

### 3.2. In Vitro Models

Immortalized cell lines grown as monolayers are often used in research, instead of primary cells. They offer several advantages, such as eliminating donor-to-donor variation, cost effectiveness, ease of use, the provision of an unlimited supply of material, and a bypassing of the ethical concerns associated with the use of animal tissue. As pure cell populations, cell lines allow for less experimental variability and greater reproducibility of results. Since their advent in the 1950s, cell lines have revolutionized scientific research and have become a central workhorse in biomedical research. However, despite being a powerful tool, one must be careful when using cell lines. Cell lines should display and maintain functional features as close to those of the primary cells as possible. This may be particularly difficult to determine, as often the functions of the primary cells are not entirely understood. Since cell lines are genetically manipulated, this may alter their phenotype, native functions, and their responsiveness to stimuli. The serial passage of cell lines can further cause genotypic and phenotypic variation over an extended period of time, and genetic drift can also cause heterogeneity in cultures at a single point in time.

#### 3.2.1. Macrophage Cell Lines

Ruminant macrophage cell lines have been widely used in order to evaluate the interaction of MAP with its target cell. BOMAC, derived from peritoneal macrophages transfected with SV40 plasmid DNA [144], is the most reported one. Although it has been argued that BOMAC cells are inherently dysfunctional, lack several receptors, are dissimilar to bovine primary-culture macrophages, and possess an insufficient capability to phagocytose MAP [145], different groups have reported comparable results regarding the MAP interaction of this cell line with respect to bovine MDM [70,146], and valuable data have been obtained with this model. Moreover, less assay-to-assay variation in the level of MAP infection in BOMAC, as compared to MDM cells, has been reported [147,148]. Studies of the metabolism and virulence of MAP have been conducted on BOMAC [148,149]. The modulation by bovine antibodies directed at mycobacterial antigens or by *Saccharomyces cerevisiae* components on BOMAC cells in response to MAP infection have been analyzed by evaluating the translocation, phagocytic activity, ROS production, immune-related gene expression, cytokine secretion, and apoptosis [148,149,150,151,152]. The use of another ruminant macrophage cell line, MOCL-4, that was established from spontaneously proliferating adherent mononuclear cells from sheep blood, has also been evaluated as a model for MAP cellular infection [149].

Murine macrophage cell lines have long been used to elucidate host–mycobacteria interactions, and the intracellular growth characteristics of MAP [54,153]. RAW 264.7 is a murine macrophage cell line which is extremely sensitive to stimulants [154]. Despite this drawback, several studies have utilized this cell line for studying nitric oxide production in response to mycobacterial infections [155,156], and have reported differences in the immune responses and gene expression elicited from the non-pathogenic and pathogenic mycobacteria [157,158]. In addition, Everman et al. [152] used this model to evaluate the effect of antibodies to opsonize MAP. Thirunavukkarasu et al. [159] also utilized RAW 264.7 cells as a model with which to examine a series of macrophage activation parameters in response to MAP, a non-pathogenic mycobacterium, as well as mycobacterial antigens. This research demonstrated the usefulness of this model for the study of mycobacterial immunopathogenesis. Moreover, this system has been featured in the selection of attenuated vaccine candidates [160]. 

The phagocytic J774.A1 is another murine macrophage cell line that has been used to evaluate macrophage activation, phagosome maturation, cytokine production, the stimulation of CD4+ T cells and the intracellular survival of MAP [55,153,161,162], and to study the antimicrobial or immunomodulatory effects of the molecules on the ability of macrophages to clear MAP infection [163,164]. 

THP-1 is a human leukemia monocytic cell line that can be differentiated into a macrophage-like phenotype using stimuli, such as phorbol-12-myristate-13-acetate (PMA) [165]. This cell line has been evaluated by several authors with the aim of shedding light on the molecular mechanisms of MAP–host interactions [166,167,168,169]. Motamedi Boroojeni et al. [170] evaluated THP-1-derived macrophages as a model with which to evaluate the immunogenicity of constructed *Salmonella*-expressing MAP-candidate vaccine genes. 

In sum, macrophage cell lines of different origins (bovine, ovine, murine, and human) have been used over time to provide valuable studies on the different aspects of the MAP–host interaction at the cellular and receptor–ligand levels, the biological differences between MAP genotypes, the importance of different bacterial virulence factors and, also, from a therapeutic perspective [164].

#### 3.2.2. Epithelial Cell Lines

The intestinal epithelium is the largest surface that acts as a primary barrier against pathogens. A mycobacterial invasion of the intestinal epithelial cells is a complex event, requiring the participation of several bacterial and host factors. Initially, the role of the microfold epithelial cells (M cells) in MAP intestinal uptake was highlighted in [171] and, afterwards, it was learned that an invasion can also occur through other cell types, in areas with or without Peyer’s patches [139], and the use of epithelial cell lines has greatly contributed to this knowledge [172,173]. 

Most of the studies on MAP and in vitro epithelial cell models have been conducted using intestinal epithelial lines of murine origin, or bovine epithelial cell lines from organs other than the intestines. Different groups have been working on obtaining bovine intestinal cell lines models [174,175]. More recently, Katwal et al. [176] developed and characterized an early passage, and immortalized a BIEC line obtained from the ileum of a 2-day-old calf. These cells expressed TLR 1–10, as well as cell-surface sugars relevant to the host–pathogen interaction [176,177]. The use of these epithelial models will surely improve the knowledge about the early interface of MAP intestinal invasion.

Additionally, some epithelial models could clarify the role of the interaction of MAP with different epithelial cell types in the intestinal mucosa. M-cell differentiation from BIECs has been achieved using a co-culture with murine Peyer’s patch lymphocytes [178] or treatment with the supernatant from bovine PBMC cultured with IL-2 [174]. In the field of mycobacterial infections, the model proposed by Kerneis et al. [178] has been used concomitantly with other mouse models to prove that the translocation by M cell is a vital entry mechanism that contributes to the pathogenesis of *Mycobacterium tuberculosis* [179]. This model would also be valuable for studying early MAP pathogenesis. Unexpectedly, despite all these reports, there is no bovine intestinal epithelial line available in culture collections. Nevertheless, other non-intestinal cell lines have been used as models for early events in a MAP invasion. Bannantine et al. [180] first used Madin–Darby bovine kidney epithelial cells (MDBK) as a model for bovine intestinal mucosa. In addition, Patel et al. [181] described that MAP was capable of infecting the confluent monolayers of MDBK cells. Since epithelial cells are not phagocytic in nature, a MAP invasion of MDBK cells indicates that the bacteria might trigger their own uptake, probably by inducing cytoskeleton reorganization [181]. This model was used to evaluate the relevance of MAP lipids and proteins on pathogenesis [182], and the role of specific antibodies in MAP invasion, translocation, and cell-mediated killing [152]. Likewise, Everman et al. [183] developed an interesting cell culture model that simulated the passage of bacteria from their uptake by the intestinal epithelium (modeled using a primary infection of MDBK monolayer infection), their spread to the tissue phagocytes (mimicked by the infection of a RAW 264.7 cell line with MAP obtained from MDBK lysates), and their ultimate return to the intestinal epithelium during the later stages of infection (modeled using a secondary infection of MDBK with MAP recovered from the intracellular compartment of the RAW 264.7). This model offered an alternative with which to study both the host and bacterial mechanisms used during an invasion, and the infection process of the ruminant intestine, and discovered unidentified changes and interactions that occur during the disease. For example, this research group performed a gene expression analysis and detected different profiles of the immune signals, as well as the different bacterial phenotypes between the primary- and secondary-infected MDBK. 

MAP has been reported to infect mammary tissue [184,185]. Various studies have demonstrated that MAP can infect the MAC-T cell line from both the apical and basolateral surfaces, with comparable efficiency and survival inside these cells, causing an impact on their gene expression [181,186]. Moreover, Patel et al. [181] described how the prior incubation of MAP in MAC-T cells enhanced the efficiency of an invasion of MDBK cells, hypothesizing that MAP present in milk could have an invasive phenotype, enhancing the infection of the suckling calves. This model was also used to evaluate the co-infection processes of MAP and other pathogens, such as *Escherichia coli*, *Staphylococcus aureus*, and *Streptococcus agalactiae*, and demonstrated the rapid baso-apical translocation of MAP in epithelial cells [187], and the fast internalization of *S. aureus* in MAC-T cells previously infected by MAP [188]. The knockout variants of TLR 4 or IL-10Rα of this model have recently been used to demonstrate the role of those receptors in the regulation of the innate response to MAP [189,190]. 

#### 3.2.3. Co-Culture of Cell Lines

The co-culture of different cell lines constitutes the simplest manner by which to include some elements of the microenvironment into the model, as an over-simplification of the complex in vivo crosstalk of the different cell types in a tissue context. Even so, such in vitro cell culture systems still offer more controllable, versatile, and reproducible setups compared to in vivo and ex vivo systems. To our knowledge, only two research studies using cell lines in co-cultures related to PTB have been published. These studies have allowed for an increased understanding of the modulation of the CD4+ T cell stimulatory capacity of MAP-infected macrophages [161], and the ability of MAP to stimulate the synthesis of integrins in macrophages, permitting their efficient attachment and translocation throughout the endothelial layer [191].

Thus far, we have reviewed and discussed the different experimental approaches used in PTB research. Table 2 summarizes the analysis carried out in this work.

## 4. Modeling PTB: Future Perspectives

The pressure to avoid animal experimentation and search for alternatives is growing. Added to this, the in vitro alternatives that are based on cell lines present limitations that have been mentioned throughout this work. In this context, other approaches are possible. The following are models that we think are worth considering in the future for PTB research.

### 4.1. Nematode Model

The nematode *Caenorhabditis elegans* can be infected by a wide variety of bacterial species, including mycobacteria, and has, therefore, been used in recent years as a model for studying some aspects of the host–pathogen interaction. Even when the absence of a clear macrophage cell lineage is posed in *C. elegans* [192], evidence from studies of different intracellular pathogens, such as *Legionella pneumophila* [193], indicates that *C. elegans*, as a host, allows for the analysis of specific virulence mechanisms which are of relevance for the pathogen–macrophage interplay. Although no study has been performed to date using this model for MAP studies, several authors have reported positive experiences using this model to evaluate other mycobacteria of pathological interest, such as *Mycobacterium avium* Subsp. *hominissuis* (MAH) [194] and *Mycobacterium marinum* [195]. Interestingly, host defense molecules, such as ROS and lysozymes, produced in response to bacteria, are delivered to the lumen of the worm intestine [196] rather than to an endosomal compartment, as would occur within macrophages. It should also be noted that the intestine of this worms is covered with an epithelium quite similar to that of the mammalian intestine, sharing a comparable morphology, structure, and function. In those experiments that evaluated *C. elegans* as a model for the pathogenesis of mycobacteria, the worms were fed with them and the infection occurred at the intestinal level. 

Galbadage et al. [197] demonstrated that *C. elegans* has particular value for studying the role of MAPK pathways in mycobacterial pathogenesis. Bermudez et al. [194] tested the pathogenicity of different adherence–deficient clones of MAH in *C. elegans*, and found that all of them had an altered ability to colonize this host. In the light of these results, *C. elegans* appears as a promising, genetically tractable pathogenicity model for both the host and the pathogen. It would be interesting to test the capacity of MAP to infect this nematode, causing a pathology, as a first step toward considering this model for pathogenesis studies, as well as for the preliminary selection of attenuated vaccine candidates for PTB.

### 4.2. Organs-on-a-Chip

Other advanced culture systems, such as organoid-derived 2D monolayer [198] and organ-on-a-chip systems, have emerged to overcome the weakness of the current organoid systems, such as their static nature, lack of media flow, and application of mechanical strain. These advanced culture platforms offer a great potential to widen the scope of research through the dissociation of 3D organoids into single cells, and their integration into advanced culture systems [199,200]. Organs-on-a-chip are self-contained and modular in vitro models with easily controllable features, involving microfluidic approaches coupled with cell culture. 

Regarding PTB, both gut-on-a-chip and immune system-on-a-chip can be attractive models. Gut-on-a-chip is fairly advanced in human medicine, and has been used to study inflammatory bowel disease [201], a gastrointestinal inflammatory disorder that shows similarity to PTB. Recent work has described the study of host–pathogen interactions in the context of intestine-on-a-chip models [202]. Studies based on human primary intestinal cells have shown that these systems have a closer transcriptomic profile and functionality to the intestine in vivo, when compared to other in vitro models. So, intestine-on-a-chip models appear as a promising tool for further research in the PTB field [134,203]. As for immune cells or immune organs-on-a-chip, these have been recently reviewed [204] and, regarding the potential application of these models in PTB, it seems that the lymph-node-on-a-chip and the inflammation-on-a-chip could be options that should be explored.

### 4.3. Precision-Cut Intestinal Slices (PCIS) 

Ruminant animal model experiments require huge budgets and the appropriate facilities, as mentioned before. Although laboratory animal models can support PTB research, sometimes the differences in results between these models and the target species can hamper translation. Added to this, there is an increasing pressure to avoid animal experimentation in any format and to search for alternatives. The in vitro alternatives based on cell lines present limitations that have been pointed out, such as lacking in vivo characteristics like fluidic flow or periodic peristalsis of limited crosstalk between the host and microorganism, and between the different cell types. In this context, there is a clear need for alternative models with higher complexity that can better resemble the disease in the natural host. As for PTB, the source of the intestine could be animals going to slaughter, taking advantage of tissue that would otherwise be residue. Precision-cut tissue slices (PCTS) should be explored, as they have been described as an invaluable model providing the cellular and structural complexity of the host tissue [205].

A type of PCTS, precision-cut intestinal slices (PCIS) have been described as a novel ex vivo model with which to study the transport, metabolism, and toxicology of drugs [206]. This technique was first reported in 2005 by de Kanter et al. [207], who used a rat small intestine and colon. The intestinal slices are obtained by embedding the excised intestinal tissue in agarose and cutting very thin sheets (200–400 μm) while it is submerged in an oxygenated buffer that guarantees cell viability, using a specialized microtome. PCIS can be incubated and remain viable for up to 24 h, permitting the study of solute transport and metabolism.

Although PCIS has been used for decades to study pharmacological and toxicological effects, the use of this technique to assess host–pathogen interactions has not been fully developed, and its literature shows far fewer studies than those using precision-cut slices of tissue from other origins, such as the lungs. Actually, precision-cut lung slices (PCLS) have been used to evaluate antimycobacterial agents against *Mycobacterium abscessus* in mice [208], and to study the early lung response to infection by *M. bovis* and *Mycobacterium tuberculosis* in cattle [209]. As of now, PCIS has been used to study viral infections, such as avian influenza [210], coronavirus [211], and hepatitis C [212], but bacterial infections have not been assessed. This is probably due to technical issues that must be resolved, and that could include the optimization of the culture medium and the development of preservation techniques that require more research, as pointed out by Li and collaborators [206].

In the near future, PCTS will be an invaluable model with which to assess host–pathogen interactions, responses to vaccines in a “whole organ” mode [205] and, hopefully, PCIS can be used to study PTB pathogenesis and evaluate vaccine candidates and/or therapies, as well. 

## 5. Conclusions

PTB control has been difficult due to the long incubation period of the disease and failures both in diagnosis and immunoprophylaxis. This review has been intended to give a view of the wide content of experimental models used to study PTB. Having standardized experimental infection models will aid in a deeper understanding of this pathology and benefit the development of control tools. So far, laboratory animal and cell-based models have contributed to an increased knowledge of MAP pathogenesis, host–MAP interaction, and the associated genetic traits. Furthermore, these models have contributed to initial phases of therapy evaluation, as well as to the testing of novel vaccine candidates. In the years to come, advances in omics approaches and their combination with the models described herein, or even with novel tissue engineering techniques, will certainly contribute to the development of new tools in the context of PTB research. 

## Figures and Tables

**Figure 1 animals-13-03553-f001:**
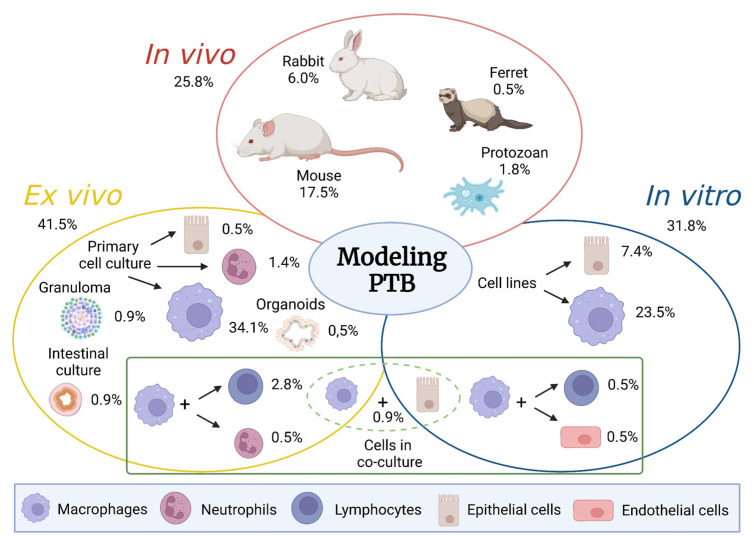
**Modeling PTB.** Relative contribution of the different models discussed in this paper. Data are expressed as the percentage of assays found for each model in PTB research. The information used to calculate these percentages was obtained by searches performed on 7 August 2023 in PubMed using the following keywords: “(mouse OR murine) AND paratuberculosis”, “rabbit AND paratuberculosis”, “(protists OR amoeba) AND paratuberculosis”, “ferret AND paratuberculosis”, “macrophage cell line AND paratuberculosis”, “macrophage cell model AND paratuberculosis”, “epithelial cells AND paratuberculosis”, “neutrophil AND paratuberculosis”, “cells co-culture AND paratuberculosis”, and ”granuloma in vitro AND paratuberculosis”. The inclusion criterion was considered to be a research study (not a review) on the use of an experimental model of infection to evaluate some aspects of PTB. In total, 217 assays were considered that met this criterion. It is important to note that these experiments were published in slightly fewer papers since, in some cases, the authors published results obtained from different models in the same paper. Since this review focuses on PTB or Johne’s disease, those works that refer to MAP but are in the context of Crohn’s disease were excluded. (Created with BioRender.com. BioRender Publication and License Rights agreement number: OB25PTVZ5Q).

**Table 1 animals-13-03553-t001:** Experimental MAP oral challenge studies in rabbits.

Age at Challenge	Infective Strain Source	Infective Dose (Dose Number)	Study Endpoint w.p.i.	Evidence of Infection % *	Experiment Objective	Reference
4–5 w	Rabbit/ hamster	1.6–5.6 mg (7)	4–40	6–50	Pathogenesis	[20]
1–2 d	Cattle	7 × 10^6^ CFU (5–10)	32–40	43	Pathogenesis	[13]
1–2 d	Cattle	3.6 × 10^8^ CFU (1)	2–36	38	Pathogenesis	[14]
1–2 d	Cattle	2.6 × 10^8^ CFU (1)	2–36	100	Pathogenesis	[14]
3 m	Cattle	5 × 10^8^ CFU (3)	104–128	50	Pathogenesis	[15]
2 w	Cattle	1 × 10^8^ CFU (3)	8–84	19	Pathogenesis	[15]
8 w	Cattle K10	1 × 10^9^ CFU (3)	16–20	40–87	Diet evaluation	[16]
16 w	Cattle K10	4 × 10^8^ CFU (3)	20	80	MAP infection and diet effect on microbiota	[23]
16 w 30 w	Cattle K10	4 × 10^8^ CFU (6)	20	100	MAP infection and diet effect on microbiota	[23]
12 w	Cattle K10	4 × 10^8^ CFU (3)	25	80	Vaccination sequence efficacy	[24]
13 w	Cattle K10	1 × 10^9^ CFU (3)	24–25	80	Vaccination routes	[27]
13–14 w	Cattle NK-764	3 × 10^8^ CFU (3)	19	60	Vaccination efficacy and trained immunity	[26]
15 w	Cattle NK-832	3 × 10^8^ CFU (3)	12	100	Effect of vaccination and probiotics and trained immunity	[22]
8–11 w	Cattle	1 × 10^9^ CFU (3)	21	100	Effect of passively transferred antibodies	[25]

w: weeks; d: days; CFU: colony-forming units; w.p.i.: weeks post-infection. * Evidence of infection was determined using post-mortem tissue culture if data were available; otherwise, MAP PCR of tissue, histology, fecal culture, or fecal PCR were considered.

**Table 2 animals-13-03553-t002:** Summary of the main experimental models for the study of PTB considered in this review.

Models		Reported Application *				References	Main Limitations	Scientific Potential
		P	D	V	T			
In vivo non-ruminant animal models	Mouse	✓		✓	✓	[4,5,6,7,8,9,10,11]	Lack of development of clinical signs.	Large amount of immunological reagents. Well-defined model. Initial vaccine screening platform.
	Rabbit	✓	✓	✓	✓	[13,14,15,16,20,22,23,24,25,26,27]	Scarcity of immunological reagents.	Clinical outcome closely resembles ruminant PTB.
	Ferret	✓	✓			[39]	Scarcity of immunological reagents. Carnivore diet. Outbred nature of the model.	Strain virulence factor research.
	Protozoa	✓				[41,43,46]	Unicellular organism.	Strain virulence factor research.
Ex vivo models	Intestinal	✓			✓	[49,50,52]	Difficult to manage. No standardized protocols available.	Early-stage pathogenesis.
	Immune cells: macrophages/ monocytes	✓		✓	✓	[33,54,55,56,57,58,59,60,61,62,63,64,65,66,67,68,69,70,71,72,73,74,75,76,78]	High variability between donors and lower reproducibility.	Strain virulence factor research. Host genetics MAP–host interaction at receptor–ligand level.
	Granulomas	✓			✓	[81,82]	Limited growth potential. Not all granuloma cell types included.	Early-stage pathogenesis. Initial vaccine screening platform.
	Immune cells: neutrophils	✓		✓		[26,93,94,95]	Role in PTB not fully understood. Short life-span cells. Inability to cryopreserve or expand them in vitro.	Early-stage host–pathogen interaction.
	Co-cultures of primary cells	✓		✓		[57,94,96,97,98,99,100,101,102,103,104,105,106]	Limited growth potential. High variability between donors.	Dissection of cross-talk between key cell types upon MAP infection. Initial vaccine screening platform.
	Organoids	✓				[119]	No standardized protocols available.	Early-stage pathogenesis.
In vitro models	Macrophage cell lines	✓		✓	✓	[54,55,69,144,145,146,147,148,149,150,151,152,153,159,161,162]	Genetically transformed cells. Lower or different response in relation to primary cells.	Strain virulence factor research. MAP–host interaction at receptor–ligand level.
	Epithelial cell lines	✓		✓	✓	[139,172,173,180,181,182,183,186,187,188,189,190]	Only used to study initial interactions as they are not the target cells.	MAP intestinal invasion (cellular and molecular level) and early pathogenesis and therapy.
	Co-culture of cell lines	✓				[161,191]	Simple and controllable setup.	Early-stage pathogenesis.

* P: pathogenesis; D: diagnosis; V: vaccination; and T: therapy.

## Data Availability

No new data were created. The results of the research engine searches run to obtain the material that has been reviewed in this work can be shared upon request to the authors.

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
