# Peer review of "Modeling Paratuberculosis in Laboratory Animals, Cells, or Tissues: A Focus on Their Applications for Pathogenesis, Diagnosis, Vaccines, and Therapy Studies"

_animals, 2023, doi:10.3390/ani13223553_

Round 1
Reviewer 1 Report
Comments and Suggestions for Authors
This is a really excellent, thorough review, very well done. I especially appreciated the (too!) brief discussion of amoeba as a model of infection, as well as the organs on a chip and precision cut slice technologies. Only two issues and some minor copyediting suggestions.
1) Please expain why (line 81) “Those works referring to MAP but in the context of Crohn’s disease were excluded.”
2) reference 141 describes research documenting that MAP invades *human* goblet cells. Do you wish to include humans in the “animal” models of MAP infection? if so, specifically say so (MAP easily infects a variety of human cell lines) and describe these human cell lines and tissues. If not, discuss why human cell lines and tissues are not appropriate.
The following are copyediting suggestions, not requirements!
1) the title, change to
vaccine and therapy studies
2) the simple summary
line 19, change to “it is considered one of the (eliminate the “as”)
line 20, change to “PTM control is still controversial”
line 21, change to “tools and vaccines have”
eliminate the last sentence of the summary, no need
3) the abstract, line 31, same change as above “tools have”
4) line 92 change to “…MAP infections become”
5) line 94 change the most near to reality to “the more realistic choice”
6) line 167 change blood sampling easiness to “ease of blood sampling”
7) line 209, eliminate the “From in vivo animal models to” so title of section reads
Ex vivo and in vitro systems
8) line 379, change limitant to “limiting factor”
9) line 707 only 1 period needeed after the word review.
10) line 734 they rather than it (they have been described)
11) line 736, change to “A type of PCTS, precision-cut…have (not has)
Comments on the Quality of English Language
very nicely written, well done, only a few trivial copyediting changes as described in my comments
Author Response
RESPONSE TO REVIEWERS animals-2582213 - Minor Revisions (Due Nov 3rd)
REVIEWER #1
This is a really excellent, thorough review, very well done. I especially appreciated the (too!) brief discussion of amoeba as a model of infection, as well as the organs on a chip and precision cut slice technologies. Only two issues and some minor copyediting suggestions.
Thank you for your comments. We have addressed your concerns, which we think have contributed to improve the review. Following is our point by point response.
1) Please explain why (line 81) “Those works referring to MAP but in the context of Crohn’s disease were excluded.”
Thank you for raising your attention on this. Our intention was to only refer to MAP infection in ruminants, this is paratuberculosis, also named Johne’s disease. We think that Crohn’s disease (CD) is an important public health issue and although MAP’s link to CD is gaining wider acceptance it is still controversial and probably deserves a separate model review to not extend this one. In any case, many of the models described in the review can be adapted to work in human disease. We have changed the original sentence to “Since this review focuses on PTB or Johne’s disease, those works referring to MAP but in the context of Crohn's disease were excluded.”(lines 88-89).
2) reference 141 describes research documenting that MAP invades *human* goblet cells. Do you wish to include humans in the “animal” models of MAP infection? if so, specifically say so (MAP easily infects a variety of human cell lines) and describe these human cell lines and tissues. If not, discuss why human cell lines and tissues are not appropriate.
This concern is somewhat related to the previous one. We did not want to give the idea that human cell lines and tissues are not appropriate at all. We do see the point that this may have been the interpretation and therefore we agree with the reviewer that if we include reference 141 since it is a human cell study, other human studies should be described. Instead replacing reference 141 we have referred to a work from Khare et al Vet Pathol 46:717–728 (2009) that was initially reference number 88 and that describes Map infection of bovine goblet cells.
Also, we have made slight changes in a paragraph of 3.2.1 section by redirecting the focus towards studies that have contributed to the knowledge of MAP-host interaction, from a paratuberculosis perspective (lines 602-608).
The following are copyediting suggestions, not requirements!
1) the title, change to
vaccine and therapy studies
Amended accordingly.
2) the simple summary
line 19, change to “it is considered one of the (eliminate the “as”)
Amended accordingly.
line 20, change to “PTM control is still controversial”
Amended accordingly.
line 21, change to “tools and vaccines have”
The Simple Summary has been modified.
eliminate the last sentence of the summary, no need
Eliminated as requested.
3) the abstract, line 31, same change as above “tools have”
Amended accordingly.
4) line 92 change to “…MAP infections become”
Amended accordingly.
5) line 94 change the most near to reality to “the more realistic choice”
Amended accordingly.
6) line 167 change blood sampling easiness to “ease of blood sampling”
Amended accordingly.
7) line 209, eliminate the “From in vivo animal models to” so title of section
Ex vivo and in vitro systems
Amended accordingly.
8) line 379, change limitant to “limiting factor”
Amended accordingly.
9) line 707 only 1 period needeed after the word review.
Amended accordingly.
10) line 734 they rather than it (they have been described)
Amended accordingly.
11) line 736, change to “A type of PCTS, precision-cut…have (not has)
Amended accordingly.

Reviewer 2 Report
Comments and Suggestions for Authors
This is very well organized and written review paper on the topic summarizing more than 200 publications. I have only minor comments as follows.
I leave the decision to the journal but the abbreviations (PTB, MAP) are not needed in the “Simple Summary” and “Abstract”.
In abstract, justifications for the use of laboratory animals and cell-based models need to be stipulated. A shorter version of lines 88-94.
In introduction (lines 44-61), it would be good to explain more about the impacts of PTB in dairy cattle and sheep.
Line 108: model not mode?
Author Response
REVIEWER #2
This is very well organized and written review paper on the topic summarizing more than 200 publications. I have only minor comments as follows.
Thank you for your comments. We have addressed your concerns, which we think have contributed to improve the review. Following is our point by point response.
I leave the decision to the journal but the abbreviations (PTB, MAP) are not needed in the “Simple Summary” and “Abstract”.
We agree with the reviewer. These abbreviations have been deleted.
In abstract, justifications for the use of laboratory animals and cell-based models need to be stipulated. A shorter version of lines 88-94.
We agree with the reviewer. The justification has been included in the abstract.
In introduction (lines 44-61), it would be good to explain more about the impacts of PTB in dairy cattle and sheep.
We agree with the reviewer. This infection has a negative impact on milk production, even in subclinical stages, as well as on body condition, as the disease progresses and intestinal malabsorption syndrome occurs. The introduction has been completed in this sense.
Line 108: model not mode?
We thank the reviewer. The word has been corrected (line 116).

Reviewer 3 Report
Comments and Suggestions for Authors
Dear Authors,
This review is describes paratuberculosis models very comprehensively. I`m impressed the figure and table.
The only weak point is too many references used - my suggestion is reduction the number of references to the most important and limit to last 20-year published.
Author Response
Dear Authors,
This review is describes paratuberculosis models very comprehensively. I`m impressed the figure and table.
The only weak point is too many references used - my suggestion is reduction the number of references to the most important and limit to last 20-year published.
Thank you. We appreciate your positive comments on the figure and table.
Regarding the number of references, it is true that we have 18 references that are over 20 years old. However, as many of them are important to situate readers and referred to a lot of important work performed prior to the year 2000, we have not considered eliminating any of them. We have eliminated instead a total of 14 other references throughout the work, which in our opinion were not strictly necessary.

Reviewer 4 Report
Comments and Suggestions for Authors
In the manuscript titled "Modeling paratuberculosis in laboratory animals, cells, or tissues: focusing on their applications for pathogenesis, diagnosis, vaccines, and therapy studies," the authors present a comprehensive overview of various models utilized in the study of paratuberculosis (PTB), encompassing both animal and cell models. The authors diligently assess the efficacy and applicability of these models, aiming to contribute significantly to the advancement of the relevant scientific community.
To enhance the quality and completeness of the manuscript, the authors should consider the following revisions:
1.Detailed Analysis of Animal Model Weaknesses:
Elaborate on the weaknesses and limitations of the animal models, particularly focusing on the ferret and protozoan models. Discuss any challenges or drawbacks related to their physiological relevance, susceptibility to disease, or limitations in accurately mimicking PTB's pathogenesis.
2. Introduction to PTB Pathogenesis:
Provide an introduction to the pathogenesis of paratuberculosis (PTB) at the beginning of the manuscript. Include key details about the etiology, transmission, and progression of the disease, highlighting the mechanisms involved in the infection and spread of Mycobacterium avium subsp. Paratuberculosis.
3. Revision of Line 147:
Remove "personal communication Fernández and Jolly" from line 147.
Comments on the Quality of English LanguageMinor editing of English language required
Author Response
REVIEWER #4
In the manuscript titled "Modeling paratuberculosis in laboratory animals, cells, or tissues: focusing on their applications for pathogenesis, diagnosis, vaccines, and therapy studies," the authors present a comprehensive overview of various models utilized in the study of paratuberculosis (PTB), encompassing both animal and cell models. The authors diligently assess the efficacy and applicability of these models, aiming to contribute significantly to the advancement of the relevant scientific community.
Thank you for your positive comments. We have addressed your concerns, which we think have contributed to improve the review. Following is our point by point response.
To enhance the quality and completeness of the manuscript, the authors should consider the following revisions:
1.Detailed Analysis of Animal Model Weaknesses:
Elaborate on the weaknesses and limitations of the animal models, particularly focusing on the ferret and protozoan models. Discuss any challenges or drawbacks related to their physiological relevance, susceptibility to disease, or limitations in accurately mimicking PTB's pathogenesis.
Weakness and limitations of such models appeared in the Table but we agree with the reviewer that this point needs to be strengthened in the text, so we have incorporated them in the lines 206-210 and 218-220, for ferret and protozoan models, respectively.
- Introduction to PTB Pathogenesis:
Provide an introduction to the pathogenesis of paratuberculosis (PTB) at the beginning of the manuscript. Include key details about the etiology, transmission, and progression of the disease, highlighting the mechanisms involved in the infection and spread of Mycobacterium avium subsp. Paratuberculosis.
We have followed the reviewer's suggestion and added a brief introduction to PTB pathogenesis in lines 45-53. We agree that this section has added completeness to the manuscript.
- Revision of Line 147:
Remove "personal communication Fernández and Jolly" from line 147.
Removed accordingly.
